# Corporate Governance Mechanisms, Ownership and Firm Value: Evidence from Listed Chinese Firms

**Yusheng Kong [1,\*], Takuriramunashe Famba [1,\*]** **, Grace Chituku-Dzimiro [2], Huaping Sun [1] and Ophias Kurauone [1]**

[1]  School of Finance & Economics, Jiangsu University, 301 Xuefu Road, Zhenjiang 212013, China;
     shp797@163.com (H.S.); Kurauoneophias@gmail.com (O.K.)
[2]  Department of Accounting & Finance, Grace Chituku-Dzimiro, Chinhoyi University of Technology,
     Chinhoyi 263, Zimbabwe; gdzimiro@gmail.com
[\*]  Correspondence: yshkong@ujs.edu.cn (Y.K.); tackrace@yahoo.com (T.F.); Tel.: +86-1826-196-8075 (T.F.)

**Abstract:** This study analyzes corporate ownership as a corporate governance mechanism and its role in creating firm value. Previous research shows that there is no convergence on the firm-value corporate ownership relationship. Most research in this area takes a cross national approach ignoring the uniqueness of each institutional setting particularly those of emerging nations. Using a unique firm level dataset, we investigate how corporate control nature and ownership concentration affect the value of Chinese listed firms. First, non-state owned control is associated with a higher Tobin's Q while a negative premium is found for state owned. Using the hybrid and the correlated random effects model we confirm a U-shaped non-linear relationship between ownership concentration and Tobin's Q, implying that firm value first decreases and then increases as block holders own more shares. Further investigation reveals that the negative effect of ownership concentration is weaker when a firm equity nature is non-state owned enterprises (non-SOEs) compared to state-owned enterprises (SOEs). While ownership concentration appears to be an efficient mechanism for corporate governance its effect is weaker for SOEs compared to non-SOEs. The results support privatization of SOEs, sound reforms such as the split share structure reform as crucial for the development of China's stock market.

**Keywords:** corporate governance; firm value; ownership concentration; block holder ownership; emerging economies; Chinese listed firms

## 1. Introduction

In the past few decades, there has been impressive body of empirical evidence and theory about the ownership of the modern public corporation. Economists have invested significantly into researching the issue of separation of ownership and control. When there is separation of ownership and control problems arise between owners and those managing their wealth (Berle and Means 1932; Jensen and Meckling 1976; Fama and Jensen 1983). This is what is known as the principal-agency conflict (Fama and Jensen 1983; Aguilera and Crespi-Cladera 2016; Zhang et al. 2016). Circumstances where interests of both parties fail to converge are fertile for principal agency conflicts. This type of conflict normally manifests itself in governance systems with widely dispersed ownership, though they can be mitigated by high levels of investor protection (Berle and Means 1932; Jensen and Meckling 1976; Shleifer and Vishny 1997). In particular if both principal and agent are utility maximizers, the agent may not always act in the interest of the principal (Jensen and Meckling 1976). As such the agent pursues objectives that are not aligned to the firm value maximization goal of shareholders for instance status, growth, permanence in the company and greater salaries (Lozano et al. 2016). This behavior is detrimental

to the shareholder's wealth, and therefore, to the firms' wealth. As such, shareholders must install mechanisms to control it.

Previous researches propose several monitoring mechanisms that shareholders can manipulate in-order to align their interests with those of managers. These mechanisms are classified into internal mechanisms, such as managerial compensation, board of directors, control by large shareholders or leverage, and external mechanisms such as the market for corporate control, the market for managers, and the market for products and services (Fama and Jensen 1983; Cuervo 2002; Becht et al. 2003; Lall 2009; Chou et al. 2011; Giroud and Mueller 2011; Fischer and Hughes 1997; Li 2014, 2019). However, the choice of mechanisms is determined by the prevalent corporate governance system in the country of interest, that is, whether it is market oriented or large shareholders oriented. When the governance system is market oriented shareholders tend to rely more on managerial compensation and the market for corporate control to solve corporate governance problems, while large shareholder oriented systems tend to use control by large incumbent shareholders to align the behavior of managers and owners (Cuervo 2002).

Firms listed in North America particularly the United States, are characterized by dispersed ownership and well-developed institutional investors (Lozano et al. 2016). While diffuse ownership structure remains debatable, economists posit that ownership structures of listed firms in some developed markets such as continental Europe and Japan, and in the Chinese framework are highly concentrated (Rong et al. 2017; Abdallah and Ismail 2017). In these jurisdictions major shareholders may be governments, families, individuals, financial institutions, and other corporations acting through a holding company or cross shareholding. In the Chinese framework, concentrated ownership structure and weak legal institutions are pervasive which creates a haven for investor expropriation (Wang 2018). Conflicts in governance systems where ownership is highly concentrated are largely between "inside" controlling shareholders and outside minority shareholders. This creates the principal/principal-agency conflict (Aguilera and Crespi-Cladera 2016). Majority shareholders may expropriate the wealth of minority shareholders at some level of ownership concentration (Shleifer and Vishny 1997).

Previous research argues that ownership structure around the world is far less dispersed than believed (La Porta et al. 1999; Holderness 2007, 2017). Furthermore the research context is, China is an emerging economy that is characterized by highly concentrated ownership structures and weak legal framework. As a result this particular research focuses on the principal/principal-agency conflict. In highly concentrated ownership environments, conflicts between owners, and managers become less important (Lozano et al. 2016).

The research aims to establish whether ownership concentration is a valuable corporate governance mechanism by examining its impact on firm value. First the study investigates how the main shareholder affect firm value for different levels of ownership. Second the study examines the impact of ownership concentration on firm value when equity natures are different. Listed firms were classified using equity nature into state-owned enterprises (SOEs) and non-state owned (non-SOEs) similar to (Rong et al. 2017). We then analyze the moderating effect of these different equity natures on the derived effects of ownership concentration on firm value in order to establish the effect of being an SOE or non-SOE on the conflict between majority and minority shareholders.

To test this hypothesis we use a sample of 1261 Chinese firms listed on the Shanghai and Shenzhen Stock Exchanges. We first measure how the main shareholder's ownership affects the value of the firm. This effect is negative for lower levels of ownership and positive for higher levels. This relationship is further tested separately for the two subsamples and the U-shaped firm-value ownership relationship is maintained for both. However an interesting discovery is that the derived inflexion points for the subsamples are different from the entire sample. The inflexion point for non-SOEs occurs at levels of ownership concentration below both the entire sample and that of SOEs. This suggests that when significant shareholders are non-SOEs they are less incentivized to expropriate minority shareholders rather they exert positive monitoring roles compared to SOEs.

To the best of our knowledge this is the first corporate governance research in the Chinese context to estimate the hybrid model (Allison 2009) augmented with the correlated random-effects model (Mundlak 1978; Wooldridge 2010) instead of the usual fixed effects model on panel data. This methodology helps us to overcome endogeneity, and unobserved heterogeneity challenges which are common in the ownership structure field. The hybrid model and the correlated random-effects models are more flexible and provide fixed-effect estimates for the level-one variables while allowing for the inclusion of level-two variables. As argued by Schunck and Perales (2017), the hybrid model and the correlated random effects model are attractive alternatives to the standard random effects and fixed effects models primarily because they differentiate within and between cluster effects and combine the strengths of random and fixed effects models. While a fixed effects (FE) model can alleviate endogeneity problems by eliminating particular time invariant, firm-specific unobservables that affect both firm value and explanatory variables (Wang 2018), it only gives the "within groups" effect estimates to time varying variables leading to misinterpretation of insignificant coefficients when confronted with an almost time invariant variable such as corporate control nature in this particular research.

Unlike other researches that are cross-national in nature this particular research controls for heterogeneity in national cultures and political institutions by focusing on a single nation (Hasan et al. 2009). For example (Lozano et al. 2016) used panel data methodology and applied the system GMM estimator on a sample of trans-national data from 16 European companies from year 2000 to 2009. Wang (2018), just like us, focused on China alone by using 6078 firm year observation from the Shanghai and Shenzhen Stock Exchanges listed firms, the research employed a random effects model that was augmented with the correlated random-effects approach (CRE). Moreover, Wang (2018) focused on examining only central state-owned firms and local state-owned firms while we extend our sample to all firms listed on the Shangai and Shenzhen Stock Exchanges. Our research, just like Lozano et al. (2016) and Wang (2018), uses market-based indicators to measure firm value contrary to the research by Yu (2013) which applied financial performance ratios. These methodological differences such as, selection of performance measures, sampling techniques, and model specifications have a propensity to yield inconsistent empirical results (Yu 2013).

We make two main theoretical contributions. First, by identifying factors that influence the function of ownership concentration as an efficient or non-efficient corporate governance mechanism, we add to the body of literature on the agency theory. Our investigation begins by looking at a framework with high ownership concentration, particularly because such an environment is characterized by high principal-principal conflicts. Our results confirm a U-shaped non-linear relationship between ownership concentration and Tobin's Q, implying that firm value first decreases and then increases as block holders own more shares. We established that the motivation to expropriate for the entire sample disappears at 58.15%. The U-shape is consistent with previous research (Wang 2018). Thus we provide support for the research work by Lozano et al. 2016 that generalizations of how ownership affects firm value from previous works based on diffusely held samples cannot be made in an environment with high ownership concentration.

Second we clarify the important role of ownership as an efficient corporate governance mechanism using the identity of the main shareholder. In particular we examine the effect of different equity natures (i.e., whether equity nature is SOEs or non-SOE). We do not attempt to establish the direct effect of equity natures but rather to determine how different equity natures influence the firm-value ownership relationship. We show how the motivation for extracting private benefits varies between SOEs and non-SOEs and that, SOEs have a higher propensity to expropriate than non-SOEs.

The second main theoretical contribution is on SOEs vs. non-SOEs business literature. We contribute to this stream of research by explaining the positive and/or the negative effects imposed by SOEs and non-SOEs firms on firm value from an agency perspective. Further investigation reveals that the negative effect of ownership concentration is weaker when a firm equity nature is non-state owned enterprises (non-SOEs) compared to state-owned enterprises (SOEs). While ownership concentration appears to be an efficient mechanism for corporate governance, its effect is weaker for

SOEs compared to non-SOEs. The computed turning points occur at 63.85% and 55.01% for SOEs and non-SOEs respectively. This shows that the negative effect of ownership on firm value is weaker when a company is a non-SOE suggesting that, the motivation to expropriate is lower for non-SOEs compared to SOEs.

To complete this paper the remainder is structured as follows: We first describe the previous corporate governance, ownership concentration, and firm value literature. We also analyze the main factors that shape the role of ownership as a good corporate governance mechanism and pose our hypotheses. We then describe data, variables, and the models and method applied empirically test the hypotheses. Following this section, we explain and discuss the results. We then conclude the paper in the final section.

## 2. Literature and Hypotheses

### 2.1. Literature

There are many studies on the separation of ownership and control of the modern corporation (Guillaume 2018; Jensen and Meckling 1979; Fama and Jensen 1983; Chen et al. 2005; Aguilera and Crespi-Cladera 2016; Mishra and Kapil 2017; Wang 2018; Hooy et al. 2019). When there is separation of ownership and control the legitimacy of corporate private property rights and of the modern corporation as an acceptable means of social control is destroyed and problems arise between owners and those managing their wealth (Berle and Means 1932; Jensen and Meckling 1976; Fama and Jensen 1983). This is what is known as the principal-agency conflict (Fama and Jensen 1983; Aguilera and Crespi-Cladera 2016; Zhang et al. 2016). Circumstances where interests of both parties fail to converge are fertile for principal agency conflicts. This type of conflict normally manifests itself in governance systems with widely dispersed ownership, though they can be mitigated by high levels of investor protection (Berle and Means 1932; Jensen and Meckling 1976; Shleifer and Vishny 1997). In particular if both principal and agent are utility maximizers, the agent may not always act in the interest of the principal (Jensen and Meckling 1976). As such the agent pursues objectives that are not aligned to the firm value maximization goal of shareholders for instance status, growth, permanence in the company and greater salaries (Lozano et al. 2016). This behavior is detrimental to the shareholder's wealth, and therefore, to the firms' wealth. As such, shareholders must install mechanisms to control it.

Previous researches propose several monitoring mechanisms that shareholders can manipulate in order to align their interests with those of managers. These mechanisms are classified into internal mechanisms, such as managerial compensation, board of directors, control by large shareholders or leverage and external mechanisms such as the market for corporate control, the market for managers, and the market for products and services (Fama and Jensen 1983; Cuervo 2002; Becht et al. 2003). Some studies have popularized the mutual monitoring mechanism in curbing agency problems (Fama and Jensen 1983; Fischer and Hughes 1997; Li 2014, 2019). Mutual monitoring can lead to better executive decisions pertaining to investment and financial policy and to a lower likelihood of unfortunate or illegal events or corporate disaster (Li 2014). A governance system that adopts mutual monitoring reduces information asymmetry between the CEO and the board. This helps to ensure that corporate fraud is prevented or detected early, excessive and short-term risk taking is discouraged while making sure that investment and financial policy decisions are made in the interest of the company (Li 2014).

Another stream of literature proposes product market competition as an effective external mechanism for disciplining management and ensuring firm performance (Cuervo 2002; Lall 2009; Chou et al. 2011; Giroud and Mueller 2011). Proponents of product market competition as an effective control mechanism argue that there are many firms that work efficiently in global markets even though they might lack effective corporate governance systems (Chou et al. 2011). When product market competition is tough, management have incentives to reduce slack and maximize profits otherwise the business will go into bankruptcy and they will be job loss (Giroud and Mueller 2011). Therefore

competition play the role of takeovers as firms with stronger management take control of the product market, and leave a much smaller share of the loser firms (Chou et al. 2011).

However, the choice of mechanisms is determined by the prevalent corporate governance system in the country of interest: that is, whether it is market oriented or large shareholders oriented. When the governance system is market oriented shareholders tend to rely more on managerial compensation and the market for corporate control to solve corporate governance problems, while large-shareholder oriented systems tend to use control by large incumbent shareholders to align the behavior of managers and owners (Cuervo 2002). Cuervo (2002) and Becht et al. (2003) opine that partial ownership and control concentration in the hands of large shareholders appears to be the most favored mechanism for collective problems among shareholders in most countries. However this form of governance creates an environment suitable for collusion of large shareholders with management against smaller investors.

Largely, previous studies on the separation of ownership and control initially focused on potential owner-manager conflicts that arise consequently from the separation of ownership and control (Monsen et al. 1968; Jensen and Meckling 1976; Demsetz 1983; Fama and Jensen 1983; Chen et al. 2005; Groß 2007; Holderness 2017; Wang 2018). This is so because many of these studies were from North America, particularly the United States where researchers argue that firms are characterized by dispersed ownership and well-developed institutional investors (Lozano et al. 2016). However there are studies that challenge this widespread diffuse ownership view in countries with high levels of investor protection (La Porta et al. 1999; Holderness 2007). These studies suggest that the diffuse ownership view in most countries including the United States is just a "myth" because corporation ownership is far less dispersed than initially thought (La Porta et al. 1999; Holderness 2007). There is a myraid of literature around the world that present evidence that most firms are characterized by high ownership concentration (La Porta et al. 1999; Holderness 2007, 2017; Lozano et al. 2016; Wang 2018).

Though the debate on whether ownership is widely diffused or highly concentrated is beyond the scope of this paper, the preceding discussion aims to contextualize and justify the significance and relevance of this research on a global scale. As discussed above, previous literature shows that ownership in firms listed in the United States and other countries with high levels of investor protection is far less diffuse than previously thought, while listed firms in China are characterized by highly concentrated ownership structures similar to firms in Europe. Ownership is mainly state ownership, institutional investors, and private individuals (Rong et al. 2017). In addition previous researches show that firms in most emerging economies are characterized by highly concentrated ownership structures and weak legal institutions which fuels investor expropriation (Céspedes et al. 2010; Wang 2018).

In such environments ownership concentration provides conditions for large shareholders to monitor the firm's management. The free-rider problem associated with dispersed ownership where no single shareholder has enough incentives to incur monitoring costs for the benefit of all shareholders. When large shareholders perform the active monitoring role, corporate decisions are better aligned with the interests of shareholders (Shleifer and Vishny 1986; Artikis et al. 2011). However ownership concentration can lead to agency problems, particularly when major shareholders acquire controlling equity holdings that tempt them to expropriate minority shareholders and deprive them of returns attributed to their investments (Shleifer and Vishny 1986). This creates the principal/principal-agency conflict that is pervasive in environments where ownership is highly concentrated and owner-manager conflicts are less important because of large shareholders' monitoring role (Lozano et al. 2016).

Our approach in this is to present corporate governance from a straightforward agency perspective similar to the work by (Shleifer and Vishny 1997; Lozano et al. 2016). This paper is guided by the view that large investors are better able to control managers' actions than small owners and recover their money (Shleifer and Vishny 1997). As such ownership concentration is presented as one of the possible corporate governance internal mechanism that efficiently solve the agency conflict between owners and managers because it solves the "free-rider problem" (Shleifer and Vishny 1986).

At some level of ownership large investors play a positive role of monitoring management. However there are some levels of ownership that result in negative effects when shareholders with

larger stakes in the business may expropriate the wealth of minority shareholders (Lozano et al. 2016). Our focus is on ownership as an internal mechanism of corporate governance in mainland China where the level of ownership concentration is high, ownership and legal protection is relatively weak, expropriating minority shareholder is lucrative and feasible for block holders who exclusively capture the whole benefit while only bearing the costs proportional to their equity positions (Wang 2018). We analyze how ownership affects the problem that exist between majority and minority shareholders, which is the most important agency relationship in cases where the level of ownership concentration is high.

Thus, higher corporate governance, considering a particular ownership structure, generates a higher performance, consequently positively affecting minority shareholders' interest. Following previous research by Yu (2013), Lozano et al. (2016), and Wang (2018), we investigate the firm value-ownership structure relationship and analyze the efficacy of the firm's ownership structure as a mechanism to enhance good corporate governance. In the next subsections we explore two factors that enhance the efficacy of ownership structure as a corporate governance mechanism: the framework and type of owner.

### 2.1.1. Framework

Ownership structures vary significantly across the world. Characteristically US firms have highly dispersed ownership structures, contrary to ownership structures of European firms and those in Mainland China which are highly concentrated (Yu 2013; Lozano et al. 2016; Wang 2018). Other researchers have however criticized the diffuse ownership view in the United States and other foreign countries (La Porta et al. 1999; Holderness 2007). According to Holderness (2017), ownership is far less diffused than thought. In the framework with highly dispersed ownership structures, ownership concentration mitigates the manager-shareholder conflicts by eliminating the "free rider" problem, and the firm value-ownership relationship yields a non-linear inverted U-shape (Shleifer and Vishny 1997). While the owner-manager conflict disappears, concentration may trigger a more complex conflict between dominant and minority shareholders primarily because the dominant shareholders have the appetite and incentive to expropriate the wealth of minority shareholders (Shleifer and Vishny 1997). To show the positive effect of monitoring, firm value increases at low ownership concentration levels then decreases at higher levels because the ability of the majority shareholder to expropriate increases with increase in control. However this relationship is valid where ownership levels are not too high up to a maximum 40% of total shares (McConnell and Servaes 1990). This is so because at such levels the benefits of expropriation are larger than the costs.

In a framework where ownership concentration is high, the "free rider" problem does not exist. The relationship is a non-linear U-shape pattern, and prior research has observed this among Chinese and European firms (Yu 2013; Lozano et al. 2016; Wang 2018). The logic is that, the main owner has enough stakes in the business to influence the company's decision, and therefore he or she is interested in the company, acts as manager or serves on the board, and has ability to expropriate. Thus the main shareholder is also an insider who may act to extract private benefits, and the motivation to do so is a function of the size of the stake he or she holds in the business. In a framework where ownership is highly concentrated, lower levels of ownership are similar to the upper levels in a dispersed framework where the relationship between ownership and firm value is negative. The motivation to expropriate disappears when the main shareholder's ownership is too high, because doing so would simply result in the direct transfer of private wealth from one venture into another, which is unlikely to benefit him except perhaps for fiscal reasons (Wang 2018).

### 2.1.2. Type of Owner

For the purpose of this research, we distinguish between SOEs and non-SOEs using equity nature of companies listed on the Shanghai and the Shenzhen Stock Exchanges. Unlike (Yu 2013; Wang 2018) who separate state-owned enterprises (SOEs) into central state-owned enterprises (CSOE),

and local state-owned enterprises (LSOE) we aggregate the two into state-owned enterprises (SOE). Our motivation to aggregate is premised on the idea that the interest and policy imperatives of both local and central government are generally similar since both are influenced by social, political, and economic conditions, and are often charged with non-financial objectives such as infrastructural financing, unemployment prevention, and welfare provision (Wang 2018). Moreover the findings by (Wang 2018) that, CSOEs are associated with higher Tobin's Q compared to LSOEs leaves us with no incentive to delve further into this line of inquiry.

Other researchers have explained the relationship between state ownership and firm value (Yu 2013; Wang 2018). To explore further this line of inquiry, we direct our efforts to understand the moderating effect of SOEs or non-SOEs on the ownership concentration and firm value relationship. We study whether ownership in the hands of SOEs or non-SOEs fosters or inhibits corporate governance. Rong et al. (2017) highlighted some of the red tape characteristics of SOEs that might increase their appetite to expropriate minority shareholders. First, listed Chinese SOEs are regarded as part of the political system and are directed by the Central Committee of the Chinese Communist Party (CCP) through a multidivisional system, which ultimately controls the mobility of government officials within it. The government has direct influence in the selection of the board chairman and CEO through the associated bureaucratic agency and the board merely rubber-stamps the decision (Rong et al. 2017). Government's direct influence on CEO appointments, ultimately exert undue influence on SOE managers to give priority to the interest of bureaucrats while minority shareholders' interests are ignored. The dilemma from this scenario is that the interests of two groups maybe conflicting: the interests of bureaucrats usually do not find convergence with the profitability goal, occasionally deviating from the goal of improving the profitability of SOEs as they focus on achieving their political goals and pursuing any private benefits (Shleifer and Vishny 1997). As noted by Sappington and Stiglitz (1987) the collision of social and political objectives with the firm's profit goals raise difficulties in management monitoring and capital budgeting apart from diluting profit making motives of local governments as corporate controllers.

We also consider the views of neoclassical economists who argue that private ownership incentivizes business to innovate and contain costs compared to public ownership (Porta et al. 1998). This view is embedded in the property rights perspective in economics (Martin and Parker 1997; Villalonga 2000) and the residual claimant theory (Rowthorn and Chang 2005). The private sector have more clearly defined property rights rather than the public sector, and thus, incentives for seeking profits by private owners which then leads to more effective management performance monitoring (Alchian 1965; McCormick and Meiners 1988). Specifically because a close linkage exists between the wealth of private controllers and the welfare of the firm, they focus more on cost minimization while maximization profits than governments (Che and Langli 2015).

Contrary to the negative effects of central government involvement in Chinese firms, government mobilizes and directs massive resources to a number of firms in key industrial sectors of the economy. CSOEs benefit in several ways among them preferential policies and treatments in areas ranging from taxation and technology transfer to material supplies and state-owned bank loans. In that vein we support that view that the central government's "helping hand" (Shleifer 1997) substitutes for weak institutional environments in factor, labor and capital markets, and provides CSOEs and their subsidiaries certain advantages that would otherwise be impossible. However as noted by (Kwon 2005), the balance between the "helping" and "grabbing" hand of a government is not clear cut but hinges on the extent to which the excessive intervention can be curtailed. The central government's as the implicit debt guarantor was effective in mitigating the financial constraints of CSOEs over the 2008 global financial crisis.

The subject of firm value ownership concentration has inspired many empirical studies. Wang (2018) in an examination of a sample of non-financial companies listed on the Shanghai and Shenzhen Stock Exchanges spanning from 2005 to 2009 found that the relationship between ownership concentration and firm value is U-Shaped. The U-shape was also established in researches

by (Anderson and Reeb 2003; Yu 2013; Lozano et al. 2016). However an inverted U-shape is established for ownership concentration levels where the main shareholder commands less than absolute control of the firm and is accompanied by a second significant Shareholder that is when the main owner can be controlled by or who colludes with another shareholder.

Inconsistent empirical results may be attributed to numerous factors such as, methodological differences, selection of performance measures, sampling technique and model specification (Yu 2013). For example, Lozano et al. (2016) used panel data methodology and applied the system GMM estimator on a sample of trans-national data from 16 European companies from year 2000 to 2009. Wang (2018) focused on China alone, using 6078 firm year observation from the Shanghai and Shenzhen Stock Exchanges listed firms employed a random effects model that was augmented with the correlated random-effects approach (CRE). Most studies have used financial ratios or market-based indicators to measure firm value. Lozano et al. (2016) used the market value of equity divided by the replacement value of total assets to define firm value, which is the main dependent variable. On another vein (Wang 2018) uses the Tobin's Q but goes further to apply an illiquidity discount of 70% based on 364 Chinese private transfers of non-tradable shares to the equity market value.

### 2.2. Hypotheses

In the Chinese framework, where ownership concentration levels are high, if the main owner effectively controls the company, a prudent strategy for increasing private benefits for shareholders when ownership increases is to do without wealth extraction but instead work on enhancing company performance. We therefore expect the main shareholder to negatively affect firm value for lower levels of ownership and positively for higher levels of ownership. Therefore the proposed hypothesis is stated as follows:

**Hypothesis 1.** *The ownership concentration-firm value relationship in the hands of the majority shareholder is negative for the lower levels of ownership concentration and positive for higher levels of concentration among Chinese firms.*

As discussed above, SOEs possess unique characteristics that do not promote good corporate governance and thus lower firm value contrary to non-SOEs. Non-SOEs do not possess some of these negative characteristics of SOEs that exacerbate agency conflicts which in-turn, have a negative impact on firm value. As a result we expect the negative impact of the main owner on firm value to be higher when company is SOEs then non-SOEs.

**Hypothesis 2.** *The negative relation between ownership concentration in the hands of the main shareholder and firm value is lower for non-SOEs than for SOEs.*

## 3. Data, Variables, Models, and Methods

### 3.1. Data

To test our hypotheses, we use a sample of non-financial firms listed on the Shanghai and Shenzhen Stock Exchanges. Implicit to Stock Exchange listed firms is that they follow rules and standards set by regulatory bodies in the course of their business operation. Moreover disclosure of financial information is standardized in line with particular authoritative accounting standards.

The time period for our analysis is from 2010 to 2016. We select this period particularly because the CSRC securities legal framework and market transparency were reviewed and significantly improved in 2009. Therefore, data from 2010 onward is relatively up-to-date, and the information disclosure requirement imposed on publicly listed firms is stricter than before that date (Lew et al. 2018).

From the population of 3556 firms across 11 industry categories (Table 1), 2295 firms were deleted. First we excluded banking and other financial institution as in Lozano et al. (2016). Second, firms that

had substantial amount of data missing were also excluded. In the end we have a sample that consist of 7727 firm-year observations, including 871 companies in 2010, 1081 in 2011, 1170 in 2012, 1168 in 2013, 1166 in 2014, 1124 in 2015, and 1147 in 2016. The firm-level financial and ownership variables are drawn from Thompson DataStream.

**Table 1.** Summarized number of companies selected by industry and percentage of representativeness. Industry classification is based on the CSRC (2001 Edition) Guidelines for Industrial Classification of Listed Companies and is compiled according to the abbreviation of Shenzhen Stock Exchange.

| Industry Code | Industry Name | Population | Deleted | Sample Size | Percentage |
|---|---|---|---|---|---|
| A | Crop Farming, Fisheries, Animal, Livestock and Forestry | 48 | 26 | 22 | 1.74% |
| B | Mining and Oil Extraction | 69 | 46 | 23 | 1.82% |
| C | Manufacturing Industry | 1573 | 735 | 838 | 66.46% |
| D | Electricity, Gas and Water | 80 | 58 | 22 | 1.74% |
| E | Civil Engineering Construction | 57 | 29 | 28 | 2.22% |
| F | Transportation Industry | 91 | 70 | 21 | 1.67% |
| G | Communication and Related Equipment Industry | 209 | 58 | 151 | 11.97% |
| H | Wholesale and Retail Industry | 135 | 77 | 58 | 4.60% |
| I | Banking, Insurance, Integrated Securities and Other Financial Institutions | 41 | 41 | 0 | 0.00% |
| J | Real Estate Development and Management Industry | 143 | 95 | 48 | 3.81% |
| K | Public Facilities | 88 | 50 | 38 | 3.01% |
| L | Information, Cultural and Communication Industry | 40 | 28 | 12 | 0.95% |
| M | Miscellaneous | 55 | 55 | 0 | 0.00% |
| N | Unnamed Industries | 927 | 927 | 0 | 0.00% |
| | **Total** | **3556** | **2295** | **1261** | **100.00%** |

*3.2. Variables and Measurement*

Variables and measurement are presented in Table 2 below

3.2.1. Dependent Variable

The dependent variable is a firm value that is defined by Durnev and Kim (2005) as an increasing function of the quality of a firm's current projects and anticipated investments within the existing corporate governance structure and institutional environment. We measure firm value using the Tobin's Q ratio, which is a hybrid performance measure that evaluates both market-based and accounting-based data. This ratio can be employed to demonstrate how a firm's shares are valued relative to a firm's property, plant (at market value), equipment, and inventory (at replacement cost) (Lozano et al. 2016). In addition the Tobin's Q complements the idea that investors favor or disfavor certain firms given the perceived risk and institutional quality and thus this will be reflected in particular Q values (Shan and McIver 2011). The Tobin's Q is also popular with current research such as Wang (2018); Jara et al. (2019); Sarhan et al. (2019); Shao (2019).

**Table 2.** Variable definition and measurement.

| Variable | Definition | Formula |
|---|---|---|
| **ROA** | Natural logarithm of Return on Assets | $\log\left(\frac{\text{net profits}}{\text{total assets}}\right)$ |
| **ASTAN** | Asset tangibility | The ration of tangible fixed assets to total assets |
| **FRM_SZ** | Firm size | The natural logarithm of total assets |
| **T'S Q** | Natural logarithm of Tobin's Q Value. | $\log\left(\frac{\text{Market Capitalisation}}{\text{total assets}}\right)$ |
| **BD_SZ** | The total number of directors on the board | Executive Directors $+$ non $-$ Executive Directors $+$ Independent non $-$ Executive Directors $+$ Chairman |
| **BD_IND** | The proportion of non-executive Directors. | $\text{BD\_IND} = \frac{\text{Total non executive directors}}{\text{Total number of directors}}$ , |
| **BD_CMTE** | Board Committee | Number of functional committee under the board |
| **BD_ACT** | Board Activism | The number of board meetings during the corresponding year |
| **OWNC_B** | Ownership concentration | The aggregate shareholding held by investors who own 5 percent or more of a firm's outstanding equity. |
| **SOE** | State-owned enterprises | SOE dummy that equals to one if the firm's equity nature is ultimately government and zero otherwise. |
| **Non-SOEs** | Non-state owned enterprises | Non-SOE dummy that equals one if the equity nature is ultimately private and zero otherwise |

### 3.2.2. Independent Variables

The main explanatory variable is ownership concentration (OWN_B), which is the percentage of shares held by the largest shareholder of the company. Following Lozano et al. (2016) and Wang (2018), ownership concentration (OWN_B) is measured by the cumulative shareholdings of block holders owning at least 5% of a firm's outstanding equity with the quadratic term ($OWN\_B^2$) detecting the possible non-linear correlation. This variable is interacted with other variables to test our hypotheses. EQN_SOE is defined as a dummy variable equal to one if the equity nature of the listed company is state-owned shares and zero otherwise. On the other hand EQN_non-SOE takes the value of one when equity nature of the listed firm is private shares and zero otherwise.

### 3.2.3. Control Variables

We introduce a variety of control variables. First, firm-specific corporate governance variables that affect firm value are controlled in order to isolate the effects of the ownership variable. From an agency theory perspective, board size (BD_SZ), board independence (BD_IND), increase monitoring of management and align their interest with those of the shareholders (Jensen and Meckling 1976). As in Wang (2018), we also include the number of functional board committees (BD_CMTE). As argued by Xie et al. (2003), board committees are tasked with monitoring and professionalizing major corporate decisions such as strategy evaluation, financial auditing, remuneration setting, and executive nomination. In addition, following the view by Vafeas (1999) that investors devalue firms with more active boards, for increased board activities signal poor performance or controversial decision-making, we include board activeness (BD_ACT) which is proxied by the frequency of the number of meetings held by the board.

Other control variables include firm size (FRM_SZ) and return on assets (ROA). FRM_SZ is a popular measure and has been used widely in corporate finance research (Shalit and Sankar 1977; Demsetz and Villalonga 2001; Sheikh et al. 2013; Dang et al. 2018; Belghitar et al. 2019). There are several firm size measures that are applied in the corporate finance domain. These include total assets, total sales, market value of equity, enterprise value (market capitalization plus debt), the number of employees, total profits, net assets (total assets minus total liabilities) (Shalit and Sankar 1977;

Dang et al. 2018). However the first three—total assets, total sales, and market value of equity—are the most popular according to Dang et al. (2018). The choice of which measure is appropriate depends on, a priori economic consideration, estimation problems, and statistical properties of various measure and the practical considerations of data availability (Shalit and Sankar 1977). This study used the total asset as a result of data availability issues and also considering that several prior studies adopted this measure (Shalit and Sankar 1977; Chou et al. 2011; Li 2014; Coles et al. 2018; Sheikh et al. 2018).

Asset tangibility (ASTAN) which is the ratio of fixed assets to total assets is also included. Tian and Estrin (2008); Irungu et al. (2018); Onguka et al. (2018), posit that firms with higher asset tangibility tend to operate in more traditional industries where growth opportunities are relatively limited.

### 3.3. Models and Methodology

Previous studies on corporate governance around the world show that there is still no consensus on the corporate governance—firm performance relation (Wintoki et al. 2012; Roberts and Whited 2013; Shao 2019). A possible cause is that some methodologies do not address endogeneity issues which are pervasive in corporate finance efficiently (Roberts and Whited 2013). Endogeneity exists when there is correlation between explanatory variables and the error term in a regression. In corporate governance research, it is not unusual to omit some variables (implicitly or explicitly) that should have been included in the vector of explanatory variables. If these omitted variables are correlated with variables already included in the vector of explanatory variables, then there is an endogeneity problem that causes inference to break down (Roberts and Whited 2013). The simultaneity condition occurs when the dependent variable ($y_{it}$), and one or more of the explanatory variable ($x_{it}$) are determined in equilibrium so that it can be plausibly argued that there is reverse causality. Researchers also use proxies in corporate finance empirical studies for unobservable or difficult to quantify variables which might lead to measurement error if there are any discrepancies that arise or if there are conceptual differences between proxies and their unobservable counterparts (Roberts and Whited 2013).

According to Roberts and Whited (2013), endogeneity leads to biased and inconsistent parameter estimates that make reliable inference virtually impossible. Wintoki et al. (2012); Roberts and Whited (2013) suggest that at least three potential sources of endogeneity exist: omitted variables, simultaneity, and measurement error. In the governance research, endogeneity can also arise from the possibility that current values of governance variables are a function of past firm performance-dynamic endogeneity (Schultz et al. 2010; Wintoki et al. 2012). As a remedy to the endogeneity problem some studies adopt the system GMM method (Holtz-Eakin et al. 1988; Arellano and Bond 1991; Arellano and Bover 1995; Blundell and Bond 1998; Wintoki et al. 2012; Roberts and Whited 2013; Zaefarian et al. 2017; Sheikh et al. 2018; Ullah et al. 2018; Shao 2019).

This study order explores the relation between ownership concentration and firm value, and we use panel data methodology to estimate the hybrid model (Allison 2009) which is augmented with the correlated random-effects model (Mundlak 1978; Wooldridge 2010). Recent research by Wang (2018) on the firm-value ownership relationship used the random effects model augmented with the correlated random effects model. The hybrid and the correlated random effects are adopted in this research because these modeling specifications are more flexible and provide fixed-effect estimates for the level-one variables and allow inclusion of level-two variables. These models permit a multilevel analysis, a methodology that retrieves effects estimates of level-two variables. Ordinarily these would be discarded in the fixed effects model, while the random effects model is less preferred in such type of research (Schunck and Perales 2017).

Previous research argues that the hybrid model and the correlated random effects model are attractive alternatives to the standard random effects and fixed effects models. These models differentiate within and between cluster effects and combine the strengths of random and fixed effects models (Schunck and Perales 2017). The fixed effects model only gives the "within groups" effect estimates

to time varying variables leading to misinterpretation of insignificant coefficients of an almost time invariant variable such as corporate control nature in this particular research. Hybrid Model:

$$FV_{it} = \alpha_0 + \alpha_1 OWNC_{it} + \alpha_2 OWNC^2_{it} + \varphi_1 CV_{it} + \varphi_2 CV_i + \delta AV_i + \mu_i + \varepsilon_{it}$$

CRE Model:

$$FV_{it} = \alpha_0 + \alpha_1 OWNC_{it} + \alpha_2 OWNC^2_{it} + \varphi_1 CV_{it} + \pi AV_i + \mu_i + \varepsilon_{it}$$

In this model FV denotes firm-value. OWNC is the variable ownership concentration while CV and AV represent control variables and averages respectively. The subscript *i* denotes level 2 (individual firms in the sample) and *t* denotes level 1 (years). Subscript *it* means the variable is level 1 and varies between and within clusters. The term $\mu_i$ is the level 2 error and the random intercept, and $\varepsilon_{it}$ is the level 1 error. The symbols $\alpha$, $\varphi$, $\delta$, are intercepts where $\delta$ is coefficient of averages less than that of *it* variable. $CV_{it}$, represent a vector of all corporate governance and financial control variables that varies between and within clusters. $CV_i$, represent a vector of all corporate governance and financial control variables that vary within clusters. $AV_i$, is the vector of the averages of all endogenous, firm specific time varying variables.

$$FV_{it} = \alpha_0 + (\alpha_1 + \gamma_1 PVT_{it})\, OWNC_{it} + (\alpha_2 + \gamma_2 PVT_{it})\, OWNC^2_{it} + \varphi_1 CV_{it} + \pi AV_i + \mu_i + \varepsilon_{it}$$

In both the hybrid and correlated random-effects models the between $\varepsilon_{it}$ and the level one covariate are restricted. This is achieved by assuming that $\boldsymbol{\varepsilon_{it}}$ depends on the mean values of the level-one covariates. In that regard we perform an estimate of the within-cluster effect, by using only within-cluster variations on the other hand between-cluster effect is estimated using between-cluster variations.

The model is estimated using the xthybrid command in stata which simplifies the specification of the hybrid as well as the correlated random effects models. As an alternative to the Hausman test, a comparison of within and between cluster effects was carried out using the Wald test to confirm the hypothesis that within-cluster effect equals between-cluster effects ($\beta_W = \beta_B$).

## 4. Results

### 4.1. Summary Statistics

A preliminary analysis of the data was conducted to ensure that there was no violation of the assumptions of normality, linearity, and multicollinearity in the sample. In Table 3 we provide the correlation matrix while Table 4 shows the summary statistics of the variables used to test the hypothesis. The independent variables show that $r < 0.7$, (Table 3) and mean VIF values 1.16, which is an assurance that there are no serious multicollinearity issues in the sample.

**Table 3.** The correlation matrix and the variance inflation factor matrix tables. Panel A: The Correlation Matrix; Panel B: The Variance Inflation Factor.

| Panel A | | | | | | | | | | |
|---|---|---|---|---|---|---|---|---|---|---|
| | ROA | BD_SZ | BD_CMTE | BD_ACT | ASTAN | FRM_SZ | BD_INDR | ONWC_B | pvtOWN~B | gvtOWN~B |
| **ROA** | 1 | | | | | | | | | |
| **BD_SZ** | −0.0074 | 1 | | | | | | | | |
| **BD_CMTE** | −0.0361 | 0.0634 | 1 | | | | | | | |
| **BD_ACT** | −0.0834 | 0.0082 | 0.0523 | 1 | | | | | | |
| **ASTAN** | −0.0531 | 0.127 | 0.0084 | −0.1428 | 1 | | | | | |
| **FRM_SZ** | −0.0913 | 0.2863 | 0.1121 | 0.2874 | 0.085 | 1 | | | | |
| **BD_INDR** | −0.0174 | −0.4736 | 0.0005 | 0.0259 | −0.0482 | −0.0208 | 1 | | | |
| **ONWC_B** | 0.0777 | −0.0515 | 0.0032 | −0.0367 | 0.0062 | 0.1139 | 0.0766 | 1 | | |
| **Non-SOEOWNC_B** | 0.1147 | −0.2398 | −0.0318 | 0.0041 | −0.174 | −0.2976 | 0.082 | 0.4203 | 1 | |
| **SOEOWNC_B** | −0.0729 | 0.2375 | 0.0469 | −0.0258 | 0.1923 | 0.4086 | −0.0314 | 0.3105 | −0.647 | 1 |

**Table 3.** *Cont.*

| Panel B | | |
|---|---|---|
| **Variable** | VIF | 1/VIF |
| **BD_SZ** | 1.45 | 0.690715 |
| **BD_INDR** | 1.31 | 0.762119 |
| **FRM_SZ** | 1.27 | 0.788465 |
| **BD_ACT** | 1.14 | 0.874277 |
| **ASTAN** | 1.05 | 0.947956 |
| **ONWC_B** | 1.04 | 0.965957 |
| **ROA** | 1.02 | 0.984303 |
| **BD_CMTE** | 1.01 | 0.986736 |
| **Mean VIF** | 1.16 | |

Table 4 provides the descriptive statistics. Panel "A" shows that the average Tobin's Q declined by a marginal 11% from 0.84 to 0.75 over the research period. Tobin's Q differs across the different classification based on equity nature. SOEs constituted 25% of our sample, private 71% while 4% was classified as other (foreign or mixed). The mean Tobin's Q values of SOEs are significantly lower than those of non-SOE, providing early evidence of the contrasting effects between government and private equity natures. On average, the companies included in the research sample have an ownership concentration of 33.25% which is high given that these companies are listed. However further classification shows that mean ownership concentration was higher for SOEs at 35.25% compared to 32.44% for non-SOEs and the difference of 2.81% is statistically significant at 5%, providing preliminary evidence that on average SOEs have higher ownership concentration compared to non-SOEs. The sample is then divided based on whether the ownership concentration degree is above or below the median (Table 5). The median of ownership concentration is the minimum percentage of ownership in the hands of the largest shareholder by one-half of the companies in the sample (Lozano et al. 2016). The mean Tobin's Q for firms with higher (upper quantile) ONW_B is 0.56 compared to 0.68 for firms with lower OWN_B (below the lower quantile). The difference of 0.12 is significant at the 5% level, indicating the adverse expropriation effect of ownership concentration.

The results show a marginal decline in mean of BD_SZ by 5% while BD_IND ratio expanded moderately by 2 to 37% which is above the (CSRC 2003) prescribed one-third requirement. As in Lew et al. 2018, we find a small standard deviation of 0.05 which reveals that the data points are quite close to the mean, indicating that most boards contain around 37% independent directors. The mean BD_ACT was 9.7. During the research period, the mean BD_ACT increased by 26%. This indicates that the board of directors increased their involvement in corporate governance over the observation period. On average, shares held by the board increased by 10% from 14.96 to 16.46.

**Table 4.** Descriptive statistics for firm value, ownership, control variables, and dummy variables.

| Panel A: Non-Dummy Variables | | | | | | | |
|---|---|---|---|---|---|---|---|
| **Control Variables** | | | | | | | |
| | **Variables** | **OBS** | **MEAN** | **SD-DEV** | **VAR** | **MIN** | **MAX** |
| **ALL YEARS** | ASTAN | 8405 | 0.21 | 0.15 | 0.02 | 0.00 | 1.00 |
| | FRM_SZ | 8405 | 21.87 | 1.21 | 1.46 | 16.71 | 27.44 |
| | ROA | 7828 | (3.55) | 0.81 | 0.66 | (12.17) | 0.15 |
| | BD_SZ | 8390 | 8.64 | 1.64 | 2.69 | 4.00 | 18.00 |
| | BD_CMTE | 8399 | 3.93 | 0.45 | 0.20 | 1.00 | 7.00 |
| | BD_ACT | 8399 | 9.70 | 4.00 | 16.00 | 1.00 | 48.00 |
| | BD_INDR | 8377 | 0.37 | 0.05 | 0.00 | 0.20 | 0.71 |
| **YR 2010** | ASTAN | 968 | 0.20 | 0.16 | 0.03 | 0.00 | 0.88 |
| | FRM_SZ | 968 | 21.53 | 1.21 | 1.46 | 16.71 | 26.69 |
| | ROA | 938 | (3.21) | 0.69 | 0.48 | (6.57) | (0.76) |
| | BD_SZ | 961 | 8.92 | 1.71 | 2.92 | 4.00 | 18.00 |
| | BD_CMTE | 965 | 3.90 | 0.51 | 0.26 | 1.00 | 7.00 |
| | BD_ACT | 968 | 8.70 | 3.62 | 13.12 | 2.00 | 38.00 |
| | BD_INDR | 959 | 0.37 | 0.05 | 0.00 | 0.20 | 0.63 |
| **YR 2016** | ASTAN | 1257 | 0.20 | 0.16 | 0.02 | 0.00 | 1.00 |
| | FRM_SZ | 1257 | 22.34 | 1.17 | 1.36 | 18.39 | 27.44 |
| | ROA | 1147 | (3.75) | 0.88 | 0.78 | (10.20) | (1.48) |
| | BD_SZ | 1257 | 8.45 | 1.64 | 2.68 | 5.00 | 18.00 |
| | BD_CMTE | 1257 | 3.95 | 0.41 | 0.17 | 2.00 | 6.00 |
| | BD_ACT | 1257 | 10.98 | 4.54 | 20.64 | 3.00 | 39.00 |
| | BD_INDR | 1257 | 0.37 | 0.05 | 0.00 | 0.20 | 0.63 |
| **Firm Value &Ownership** | | | | | | | |
| **All YEARS** | T'SQ | 8326 | 0.62 | 0.80 | 0.63 | (2.50) | 3.93 |
| | ONWC_B | 8405 | 0.33 | 0.14 | 0.02 | 0.04 | 0.89 |
| **YR 2010** | T'SQ | 912 | 0.84 | 0.71 | 0.50 | (1.79) | 2.74 |
| | ONWC_B | 968 | 0.35 | 0.15 | 0.02 | 0.04 | 0.89 |
| **YR 2016** | T'SQ | 1252 | 0.75 | 0.82 | 0.67 | (2.01) | 3.93 |
| | ONWC_B | 1257 | 0.30 | 0.13 | 0.02 | 0.06 | 0.82 |
| **Panel B** | | | | | | | |
| **Dummy Variable: Equity Nature** | | | | | | | |
| | **Variable** | **Mean** | **Std. Dev.** | **Freq.** | **PER** | | |
| **All YEAR** | EQN_non-SOE | 0.80 | 0.69 | 6131 | 74 | | |
| | EQN_SOE | 0.11 | 0.86 | 2195 | 26 | | |
| | EQN_SOE | 0.11 | 0.86 | 2195 | 26 | | |
| **YR 2010** | EQN_non-SOE | 1.06 | 0.56 | 603 | 66 | | |
| | EQN_SOE | 0.41 | 0.76 | 309 | 34 | | |
| **YR 2016** | EQN_non-SOE | 0.95 | 0.69 | 944 | 75 | | |
| | EQN_SOE | 0.14 | 0.87 | 308 | 25 | | |
| **Dummy Variable: Industry** | | | | | | | |
| **All YEAR** | NON_MAN | 0.56 | 0.92 | 2763 | 33 | | |
| | MAN | 0.65 | 0.72 | 5563 | 67 | | |
| **YR 2010** | NON_MAN | 0.76 | 0.81 | 311 | 34 | | |
| | MAN | 0.88 | 0.65 | 601 | 66 | | |
| **YR 2016** | NON_MAN | 0.63 | 0.94 | 415 | 33 | | |

**Table 5.** Results from univariate tests.

| Panel A: Mean Comparison | | | |
|---|---|---|---|
| | | T'S Q | |
| | OBS | Mean | Difference |
| MAN | 5563 | 0.560 | |
| NON_MAN | 2763 | 0.649 | 0.089 *** |
| EQN_non-SOE | 6131 | 0.800 | |
| EQN_SOE | 2195 | 0.114 | 0.686 *** |
| Upper Quantile | | | |
| EQN_non-SOE | 3519 | 0.820 | |
| EQN_SOE | 1067 | 0.179 | 0.641 *** |
| Lower Quantile | | | |
| EQN_non-SOE | 2611 | 0.773 | |
| EQN_SOE | 1129 | 0.053 | 0.720 *** |

| Panel B: Median Comparison | | | | |
|---|---|---|---|---|
| | | T's Q | | |
| Below median | OBS | Median | Mean | |
| EQN_non-SOE | 2543 | 0.817 | 0.166 | |
| EQN_SOE | 1620 | 0.129 | (0.261) | 0.426 *** |
| Above median | | | | |
| EQN_non-SOE | 3587 | 0.817 | 1.250 | |
| EQN_SOE | 576 | 0.129 | 1.170 | 0.081 *** |

| | OWN_CONC | | |
|---|---|---|---|
| | OBS | Mean | Difference |
| SOE | 2200 | 0.352 | |
| Non-SOE | 6205 | 0.325 | 0.027 *** |
| MAN | 5612 | 0.334 | |
| NON_MAN | 2793 | 0.329 | 0.005 |
| OWN | | | |
| Below median | | | |
| EQN_non-SOE | 3234 | 0.212 | |
| EQN_SOE | 965 | 0.220 | 0.001 |
| Above median | | | |
| EQN_non-SOE | 2967 | 0.441 | |
| EQN_SOE | 1235 | 0.456 | 0.015 *** |

Note: The mean and median comparisons use the unpaired T-test and the Wilconxon Z-test respectively, *** $p < 0.01$.

## 4.2. Regression Results

The regression results are presented in Tables 6 and 7. Table 6 presents the estimation using the hybrid model while Table 7 uses the correlated random effects model. First we conducted the baseline estimation (Column 1 of Tables 6 and 7) capturing only financial and corporate governance control variables. Second we include ownership concentration (OWNC_B) in model 2 column 2, Tables 6 and 7. The last two columns capture the interaction variables non-SOEOWNC_B and SOEOWNC_B respectively.

**Table 6.** Tobin's Q on Equity Nature and Ownership Concentration application of the Hybrid Model and the xthybrid command in Stata on the entire sample.

| Variable | Model 1 | Model 2 | Model 3 | Model 4 |
|---|---|---|---|---|
| W__ONWC_B | | −2.591 *** | −3.205 *** | −2.259 *** |
| | | (0.3500) | (0.4050) | (0.3640) |
| W__ONWC_B2 | | 2.228 *** | 2.913 *** | 1.769 *** |
| | | (0.4400) | (0.5560) | (0.4750) |
| W__FRM_SZ | −0.0377 *** | −0.0695 *** | −0.0685 *** | −0.0689 *** |
| | (0.0111) | (0.0116) | (0.0116) | (0.0116) |
| W__ASTAN | −0.358 *** | −0.365 *** | −0.358 *** | −0.357 *** |
| | (0.0741) | (0.0736) | (0.0736) | (0.0736) |
| W__ROA | 0.137 *** | 0.146 *** | 0.145 *** | 0.145 *** |
| | (0.0080) | (0.0080) | (0.0080) | (0.0080) |
| W__BD_SZ | −0.0279 *** | −0.0267 *** | −0.0251 *** | −0.0241 *** |
| | (0.0068) | (0.0068) | (0.0068) | (0.0068) |
| W__BD_CMTE | (0.0227) | (0.0221) | (0.0213) | (0.0222) |
| | (0.0204) | (0.0202) | (0.0202) | (0.0202) |
| W__BD_ACT | 0.0180 *** | 0.0183 *** | 0.0182 *** | 0.0181 *** |
| | (0.0018) | (0.0018) | (0.0018) | (0.0018) |
| W__BD_INDR | (0.1890) | (0.1900) | (0.2020) | (0.2080) |
| | (0.1730) | (0.1720) | (0.1710) | (0.1710) |
| B__ONWC_B | | −0.724* | −1.025 *** | (0.3850) |
| | | (0.3860) | (0.3950) | (0.4090) |
| B__ONWC_B2 | | 1.094** | 1.607 *** | 0.5830 |
| | | (0.5110) | (0.5500) | (0.5650) |
| B__FRM_SZ | −0.450 *** | −0.456 *** | −0.444 *** | −0.445 *** |
| | (0.0112) | (0.0116) | (0.0123) | (0.0123) |
| B__ASTAN | −0.884 *** | −0.882 *** | −0.852 *** | −0.852 *** |
| | (0.0822) | (0.0820) | (0.0824) | (0.0825) |
| B__ROA | 0.398 *** | 0.398 *** | 0.390 *** | 0.391 *** |
| | (0.0186) | (0.0186) | (0.0187) | (0.0187) |
| B__BD_SZ | 0.0023 | 0.0026 | 0.0067 | 0.0068 |
| | (0.0092) | (0.0093) | (0.0093) | (0.0093) |
| B__BD_CMTE | (0.0430) | (0.0420) | (0.0421) | (0.0420) |
| | (0.0280) | (0.0280) | (0.0278) | (0.0279) |
| B__BD_ACT | 0.0063 | 0.00694* | 0.0057 | 0.0059 |
| | (0.0039) | (0.0039) | (0.0039) | (0.0039) |
| B__BD_INDR | 0.875 *** | 0.838 *** | 0.833 *** | 0.828 *** |
| | (0.2740) | (0.2740) | (0.2730) | (0.2730) |
| W__non-SOEOWNC_B | | | 0.822 *** | |
| | | | (0.2940) | |
| W__non-SOEOWNC_B2 | | | −0.949 * | |
| | | | (0.5270) | |
| B__non-SOEOWNC_B | | | 0.681 *** | |
| | | | (0.2350) | |
| B__SOEOWNC_B2 | | | −1.085 ** | |
| | | | (0.4860) | |
| W__SOEOWNC_B | | | | −1.450 *** |
| | | | | (0.3980) |
| W__SOEOWNC_B2 | | | | 1.890 *** |
| | | | | (0.6150) |
| B__SOEOWNC_B | | | | −0.663 *** |
| | | | | (0.2430) |
| B__SOEOWNC_B2 | | | | 1.070** |
| | | | | (0.4990) |
| Constant | 11.85 *** | 12.07 *** | 11.73 *** | 11.76 *** |
| | (0.2610) | (0.2790) | (0.3000) | (0.3030) |
| Observations | 7717 | 7717 | 7717 | 7717 |
| Number of firms | 1262 | 1262 | 1262 | 1262 |

This table reports the results of the hybrid model of firm value on, ownership concentration, control variables and dummy variables generated using the xthybrid command in Stata. Model 2 is an extension of Model 1 by adding ownership concentration, while models 3 and 4 are incremental to model 2 and capture the interaction between ownership concentration and equity nature, non-SOE, or SOE respectively. The sample period is from 2010 to 2016. The t-statistics are in parentheses while ***, ** and * statistical significance at the 1%, 5%, and 10% respectively.

**Table 7.** Tobin's Q on Equity Nature and Ownership Concentration using the correlated random effects model (CREM) and the xthybrid command in Stata on the entire sample.

| VARIABLES | Model 1 | Model 2 | Model 3 | Model 4 |
|---|---|---|---|---|
| W__ASTAN | −0.358 *** | −0.365 *** | −0.358 *** | −0.357 *** |
|  | (0.0741) | (0.0736) | (0.0736) | (0.0736) |
| W__FRM_SZ | −0.0377 *** | −0.0695 *** | −0.0685 *** | −0.0689 *** |
|  | (0.0111) | (0.0116) | (0.0116) | (0.0116) |
| W__BD_INDR | (0.1890) | (0.1900) | (0.2020) | (0.2080) |
|  | (0.1730) | (0.1720) | (0.1710) | (0.1710) |
| W__ROA | 0.137 *** | 0.146 *** | 0.145 *** | 0.145 *** |
|  | (0.0080) | (0.0080) | (0.0080) | (0.0080) |
| W__BD_SZ | −0.0279 *** | −0.0267 *** | −0.0251 *** | −0.0241 *** |
|  | (0.0068) | (0.0068) | (0.0068) | (0.0068) |
| W__BD_CMTE | (0.0227) | (0.0221) | (0.0213) | (0.0222) |
|  | (0.0204) | (0.0202) | (0.0202) | (0.0202) |
| W__BD_ACT | 0.0180 *** | 0.0183 *** | 0.0182 *** | 0.0181 *** |
|  | (0.0018) | (0.0018) | (0.0018) | (0.0018) |
| W__ONWC_B |  | −2.591 *** | −3.205 *** | −2.259 *** |
|  |  | (0.3500) | (0.4050) | (0.3640) |
| W__ONWC_B2 |  | 2.228 *** | 2.913 *** | 1.769 *** |
|  |  | (0.4400) | (0.5560) | (0.4750) |
| W__non-SOEOWNC_B |  |  | 0.822 *** |  |
|  |  |  | (0.2940) |  |
| W__non-SOEOWNC_B2 |  |  | −0.949 * |  |
|  |  |  | (0.5270) |  |
| D__ONWC_B |  | 1.866 *** | 2.180 *** | 1.874 *** |
|  |  | (0.5210) | (0.5660) | (0.5480) |
| D__ONWC_B2 |  | −1.135 * | −1.306 * | (1.1860) |
|  |  | (0.6740) | (0.7820) | (0.7380) |
| D__ASTAN | −0.525 *** | −0.517 *** | −0.494 *** | −0.495 *** |
|  | (0.1110) | (0.1100) | (0.1100) | (0.1110) |
| D__FRM_SZ | −0.412 *** | −0.386 *** | −0.375 *** | −0.376 *** |
|  | (0.0158) | (0.0164) | (0.0169) | (0.0169) |
| D__ROA | 0.260 *** | 0.253 *** | 0.245 *** | 0.246 *** |
|  | (0.0203) | (0.0203) | (0.0203) | (0.0203) |
| D__BD_SZ | 0.0303 *** | 0.0293 ** | 0.0317 *** | 0.0309 *** |
|  | (0.0115) | (0.0115) | (0.0115) | (0.0116) |
| D__BD_CMTE | (0.0203) | (0.0199) | (0.0208) | (0.0198) |
|  | (0.0347) | (0.0345) | (0.0344) | (0.0344) |
| D__BD_ACT | −0.0117 *** | −0.0114 *** | −0.0126 *** | −0.0122 *** |
|  | (0.0043) | (0.0043) | (0.0043) | (0.0043) |
| D__BD_INDR | 1.064 *** | 1.028 *** | 1.035 *** | 1.035 *** |
|  | (0.3240) | (0.3230) | (0.3220) | (0.3230) |
| D__non-SOEOWNC_B |  |  | (0.1410) |  |
|  |  |  | (0.3760) |  |
| D__non-SOEOWNC_B2 |  |  | (0.1360) |  |
|  |  |  | (0.7170) |  |
| W__non-SOEOWNC_B2 |  |  |  | 1.890 *** |
|  |  |  |  | (0.6150) |
| W__SOEOWNC_B |  |  |  | −1.450 *** |
|  |  |  |  | (0.3980) |
| D__SOEOWNC_B |  |  |  | 0.787 * |
|  |  |  |  | (0.4660) |
| D__SOEOWNC_B2 |  |  |  | (0.8200) |
|  |  |  |  | (0.7920) |
| Constant | 11.85 *** | 12.07 *** | 11.73 *** | 11.76 *** |
|  | (0.2610) | (0.2790) | (0.3000) | (0.3030) |
| Observations | 7717 | 7717 | 7717 | 7717 |
| Number of groups | 1262 | 1262 | 1262 | 1262 |

This table reports the results of the correlated random effects model of firm value on, ownership concentration, control variables, and dummy variables generated using the xthybrid command in Stata. Model 2 is an extension of model 1 by adding ownership concentration, while models 3 and 4 are incremental to model 2 and capture the interaction between ownership concentration and equity nature, non-SOE, or SOE respectively. The t-statistics are in parentheses while ***, ** and * statistical significance at the 1%, 5%, and 10% respectively.

The results show that the within cluster coefficients generated using both methodology are similar. In the hybrid model, the between-cluster effects and within cluster effects were found to be statistically significantly different from each other, thus rejecting the standard random-intercept model as the appropriate model in favor of the hybrid and correlated radon effects. Substantively, this means that the random effects model's assumption of a zero correlation between the level-two error and level-one covariates does not hold. In the baseline estimation (column 1, Tables 6 and 7), the coefficient of within cluster BD_SZ is negative and has statistical economic significance suggesting that a larger board typically incurs higher coordinating costs and entrenchment risk, consistent with the findings by Nguyen et al. (2016) and Wang (2018). The within cluster coefficient of BD_ACT is positive and is statistically significant. This result suggests a possible significant role of the board. Wang (2018) posit that increased supervisory board activities improve monitoring and decision-making. A negative significant coefficient for ASTAN consistent with Wang (2018) was established. Firms with higher ASTAN, operate in more traditional industries where growth opportunities are relatively limited which might negatively impact firm values.

Model 2, Column 2 of Table 6, reveals the negative effect at lower levels of ownership concentration in the hands of the main shareholder on firm value and a positive effect at some levels of this ownership concentration. The results show that the coefficient of OWNC_B is negative, ($\alpha 1 = -2.591$, p < 0.0001) at the lower levels of ownership and that the coefficient of quadratic term OWNC_B2 is positive, $\alpha_2 = 2.228$, $p < 0.0001$, at the higher levels of ownership. The signs and magnitudes of the coefficient on the coefficients indicate a nonlinear correlation between firm-value and block holder ownership (Wang 2018). The result confirms Hypothesis 1. Following Lozano et al. (2016), we calculated the turning point to establish the point for which the motivation to expropriate disappears. This was achieved by running the first derivative of the firm value-ownership equation, and equating the gradient function to zero-$\left(\frac{dFV}{dCONC\_B} = 0\right)$. We established that the motivation to expropriate for the entire sample disappears at 58.15%. Further running the second derivative of our firm value-ownership equation-$\left(\frac{d^2FV}{d^2CONC\_B}\right)$ we obtain a positive sign to confirm that the value obtained is a minimum point. As in Lozano et al. (2016); Wang (2018), we do not find the free rider problem among Chinese listed firms because of the high level ownership concentration of the controlling shareholder. This provides support for the view that, expropriation incentives for the main shareholder are higher at lower levels of ownership, because the costs to do so are lower than the benefits. These incentives, at some levels of ownership, are outweighed by the costs causing the extraction of private benefits to disappear, at which point the monitoring function overcomes collusion.

Column 3 and 4, Table 6 present the results that include cross variable non-SOEOWNC_B and SOEOWNC_B to moderate the ownership firm value relation. In this model, we examine how equity nature differentiates the expropriation effect. In that seam, we determine the minimum turning point where the incentive to expropriate disappears. We find a U-shaped relation between ownership and firm-value for both SOEs and non-SOEs. For non-SOEs the coefficient, ($\alpha_1 = -3.205$, $p < 0.0001$ and, $\alpha_2 = 2.913$, $p < 0.0001$ while for the SOEs, ($\alpha_1 = -2.259$, $p < 0.0001$ and, $\alpha_2 = 1.769$, $p < 0.0001$. The computed turning points occur at 55.01% and 63.85% for non-SOEs and SOEs respectively. This shows that the negative effect of ownership on firm value is weaker when a company is a non-SOE suggesting that, the motivation to expropriate is lower for non-SOEs compared to SOEs. On the relationship between firm value and block holder ownership, our results consistently mirror the finding by Lozano et al. (2016); Wang (2018). Similar to Lozano et al. (2016) and Wang (2018), we found a U-shaped relationship between ownership concentration and firm value. However our results show different inflexion points. While we unearth the inflexion point at around 58.15%, Lozano et al. (2016) found that the inflexion point for companies with lower investor protection occurs at around 79%, which is in the region of the 75% discovered by Wang (2018). These differences can be traced to sampling differences as evidenced by the mean of OWNC_B. While the mean ownership concentration used in this study is 33.35%, Wang (2018) uses a sample made up of only state-owned companies with

mean ownership concentration of 46.9%. Moreover Lozano et al. (2016) used a sample from European firms only.

### 4.3. Robustness Checks

As the research sample shows that manufacturing firms were 66.46% of the sample, we carry out further robust tests, splitting the entire sample into two sub-samples, manufacturing firms and non-manufacturing firms. The results of the estimated hybrid models for the two different samples are presented in Tables 8 and 9. Largely, the robust test results reported in Table 8, replicate results presented in Table 7. The coefficients of ONWC_B and the quadratic term OWNC_B2 retained the same negative and positive signs respectively though the magnitudes of the coefficients are different. Perhaps the major notable difference is on the within cluster effect of variable FRM_SZ which loses statistical significance though the coefficients largely retain the negative sign.

**Table 8.** Tobin's Q on Equity Nature and Ownership Concentration using the Hybrid Model and the xthybrid command in Stata on the sample of manufacturing companies.

| VARIABLES | Model 1 | Model 2 | Model 3 | Model 4 |
|---|---|---|---|---|
| **W__ONWC_B** | | −1.993 *** | −2.871 *** | −1.435 *** |
| | | (0.4340) | (0.4930) | (0.4460) |
| **W__ONWC_B2** | | 1.674 *** | 2.967 *** | 0.8220 |
| | | (0.5610) | (0.7030) | (0.5890) |
| **W__FRM_SZ** | 0.0074 | (0.0197) | (0.0194) | (0.0190) |
| | (0.0139) | (0.0145) | (0.0145) | (0.0144) |
| **W__ASTAN** | −0.358 *** | −0.363 *** | −0.357 *** | −0.352 *** |
| | (0.0836) | (0.0835) | (0.0835) | (0.0833) |
| **W__ROA** | 0.132 *** | 0.139 *** | 0.139 *** | 0.139 *** |
| | (0.0099) | (0.0099) | (0.0099) | (0.0099) |
| **W__BD_SZ** | −0.0224 *** | −0.0204** | −0.0183** | −0.0169** |
| | (0.0082) | (0.0082) | (0.0082) | (0.0082) |
| **W__BD_CMTE** | (0.0024) | 0.0011 | 0.0006 | 0.0015 |
| | (0.0241) | (0.0240) | (0.0240) | (0.0239) |
| **W__BD_ACT** | 0.0211 *** | 0.0213 *** | 0.0211 *** | 0.0208 *** |
| | (0.0023) | (0.0023) | (0.0023) | (0.0023) |
| **W__BD_INDR** | 0.0025 | (0.0067) | (0.0520) | (0.0635) |
| | (0.2120) | (0.2110) | (0.2110) | (0.2100) |
| **W__non-SOEOWNC_B** | | | 1.274 *** | |
| | | | (0.3490) | |
| **W__non-SOEOWNC_B2** | | | −1.890 *** | |
| | | | (0.6550) | |
| **B__ONWC_B** | | (0.5640) | (0.5370) | (0.5920) |
| | | (0.4260) | (0.4380) | (0.4570) |
| **B__ONWC_B2** | | 0.975 * | 0.9300 | 1.0000 |
| | | (0.5590) | (0.5970) | (0.6280) |
| **B__FRM_SZ** | −0.405 *** | −0.412 *** | −0.413 *** | −0.416 *** |
| | (0.0133) | (0.0136) | (0.0144) | (0.0145) |
| **B__ASTAN** | −0.985 *** | −0.981 *** | −0.982 *** | −0.988 *** |
| | (0.1060) | (0.1060) | (0.1060) | (0.1060) |
| **B__ROA** | 0.395 *** | 0.394 *** | 0.395 *** | 0.396 *** |
| | (0.0207) | (0.0206) | (0.0209) | (0.0209) |
| **B__BD_SZ** | (0.0066) | (0.0051) | (0.0056) | (0.0066) |
| | (0.0105) | (0.0105) | (0.0106) | (0.0107) |
| **B__BD_CMTE** | (0.0342) | (0.0345) | (0.0345) | (0.0341) |
| | (0.0340) | (0.0338) | (0.0339) | (0.0338) |
| **B__BD_ACT** | 0.0132 ** | 0.0141 *** | 0.0143 *** | 0.0147 *** |
| | (0.0052) | (0.0052) | (0.0052) | (0.0052) |
| **B__BD_INDR** | 0.803 *** | 0.775 *** | 0.775 *** | 0.763 ** |
| | (0.2970) | (0.2960) | (0.2960) | (0.2970) |
| **B__non-SOEOWNC_B** | | | (0.0599) | |
| | | | (0.2600) | |
| **B__non-SOEOWNC_B2** | | | 0.0945 | |
| | | | (0.5390) | |

**Table 8.** *Cont.*

| VARIABLES | Model 1 | Model 2 | Model 3 | Model 4 |
|---|---|---|---|---|
| W__SOEOWNC_B | | | | −2.446 *** |
| | | | | (0.4750) |
| W__SOEOWNC_B2 | | | | 3.857 *** |
| | | | | (0.7880) |
| B__SOEOWNC_B | | | | 0.1040 |
| | | | | (0.2720) |
| B__SOEOWNC_B2 | | | | (0.0874) |
| | | | | (0.5560) |
| Constant | 10.87 *** | 11.08 *** | 11.11 *** | 11.18 *** |
| | (0.3090) | (0.3280) | (0.3510) | (0.3560) |
| Observations | 5199 | 5199 | 5199 | 5199 |
| Number of groups | 840 | 840 | 840 | 840 |

This table reports the results of the sub-sample of manufacturing firms. We estimate the hybrid model of firm value on, ownership concentration, control variables, and dummy variables generated using the xthybrid command in Stata. Model 2 is an extension of model 1 by adding ownership concentration, while model 3 and 4 are incremental to model 2 and capture the interaction between ownership concentration and equity nature, non-SOE or SOE respectively. The sample period is from 2010 to 2016. The t-statistics are in parentheses while ***, ** and * statistical significance at the 1%, 5%, and 10% respectively.

**Table 9.** Tobin's Q on Equity Nature and Ownership Concentration using the hybrid model and the xthybrid command in Stata on the sample of non-manufacturing companies.

| | Model 1 | Model 2 | Model 3 | Model 4 |
|---|---|---|---|---|
| W__ONWC_B | | −3.655 *** | −3.731 *** | −3.862 *** |
| | | (0.6060) | (0.7540) | (0.6380) |
| W__ONWC_B2 | | 3.011 *** | 2.685 *** | 3.431 *** |
| | | (0.7230) | (0.9380) | (0.8060) |
| W__FRM_SZ | −0.106 *** | −0.149 *** | −0.147 *** | −0.147 *** |
| | (0.0186) | (0.0195) | (0.0195) | (0.0195) |
| W__ASTAN | −0.429 *** | −0.430 *** | −0.422 *** | −0.424 *** |
| | (0.1570) | (0.1550) | (0.1550) | (0.1550) |
| W__ROA | 0.147 *** | 0.157 *** | 0.157 *** | 0.157 *** |
| | (0.0136) | (0.0134) | (0.0134) | (0.0134) |
| W__BD_SZ | −0.0306 ** | −0.0331 *** | −0.0332 *** | −0.0335 *** |
| | (0.0123) | (0.0121) | (0.0121) | (0.0122) |
| W__BD_CMTE | −0.0774 ** | −0.0873 ** | −0.0869 ** | −0.0872 ** |
| | (0.0377) | (0.0372) | (0.0372) | (0.0372) |
| W__BD_ACT | 0.0157 *** | 0.0162 *** | 0.0163 *** | 0.0163 *** |
| | (0.0028) | (0.0028) | (0.0028) | (0.0028) |
| W__BD_INDR | (0.4860) | (0.4580) | (0.4410) | (0.4550) |
| | (0.2970) | (0.2930) | (0.2940) | (0.2930) |
| W__non−SOEOWNC_B | | | (0.0493) | |
| | | | (0.5690) | |
| W__non−SOEOWNC_B2 | | | 0.6400 | |
| | | | (0.8990) | |
| B__ONWC_B | | (0.7620) | −1.556 ** | 0.3700 |
| | | (0.7630) | (0.7690) | (0.7850) |
| B__ONWC_B2 | | 1.0280 | 2.354 ** | (0.5600) |
| | | (1.0220) | (1.1110) | (1.0920) |
| B__FRM_SZ | −0.513 *** | −0.515 *** | −0.477 *** | −0.481 *** |
| | (0.0203) | (0.0211) | (0.0220) | (0.0219) |
| B__ASTAN | −0.836 *** | −0.835 *** | −0.648 *** | −0.654 *** |
| | (0.1430) | (0.1430) | (0.1430) | (0.1440) |
| B__ROA | 0.384 *** | 0.386 *** | 0.367 *** | 0.370 *** |
| | (0.0377) | (0.0377) | (0.0368) | (0.0368) |
| B__BD_SZ | 0.0110 | 0.0099 | 0.0176 | 0.0179 |
| | (0.0173) | (0.0174) | | (0.0170) |
| B__BD_CMTE | (0.0389) | (0.0364) | (0.0387) | (0.0293) |
| | (0.0481) | (0.0481) | (0.0466) | (0.0468) |

**Table 9.** *Cont.*

|  | Model 1 | Model 2 | Model 3 | Model 4 |
|---|---|---|---|---|
| **B__BD_ACT** | 0.0018 | 0.0020 | (0.0009) | (0.0006) |
|  | (0.0063) | (0.0063) | (0.0063) | (0.0063) |
| **B__BD_INDR** | 0.8950 | 0.8540 | 0.8170 | 0.7060 |
|  | (0.5590) | (0.5630) | (0.5460) | (0.5480) |
| **B__non−SOEOWNC_B** |  |  | 1.964 *** |  |
|  |  |  | (0.4590) |  |
| **B__non−SOEOWNC_B2** |  |  | −3.009 *** |  |
|  |  |  | (0.9440) |  |
| **W__SOEOWNC_B** |  |  |  | 0.2770 |
|  |  |  |  | (0.7720) |
| **W__SOEOWNC_B2** |  |  |  | (0.9380) |
|  |  |  |  | (1.0340) |
| **B__SOEOWNC_B** |  |  |  | −1.922 *** |
|  |  |  |  | (0.4700) |
| **B__SOEOWNC_B2** |  |  |  | 2.957 *** |
|  |  |  |  | (0.9630) |
| **Constant** | 13.14 *** | 13.32 *** | 12.28 *** | 12.37 *** |
|  | (0.4760) | (0.5160) | (0.5440) | (0.5410) |
| **Observations** | 2518 | 2518 | 2518 | 2518 |
| **Number of groups** | 422 | 422 | 422 | 422 |

This table reports the results of the sub-sample of non-manufacturing firms. We estimate the hybrid model of firm value on, ownership concentration, control variables, and dummy variables generated using the xthybrid command in Stata. Model 2 is an extension of model 1 by adding ownership concentration, while model 3 and 4 are incremental to model 2 and capture the interaction between ownership concentration and equity nature, non-SOE or SOE respectively. The sample period is from 2010 to 2016. The t-statistics are in parentheses while *** and ** statistical significance at the 1% and 5%.

We further analyze the turning points. The robustness check results show that the turning point is substantially similar at 59.65% for 66.46% of the manufacturing firms compared to 58.15% for the entire sample. The results also confirm the findings that SOEs have a higher propensity to expropriate compared to SOEs. The turning points for are 48.22% and 87.29% for non-SOEs and SOEs respectively.

## 5. Conclusions

The focus of this study is to explore the principal-principal agency relationship. To delve into this relationship, we measure the effect of the largest shareholder on firm value. Two important variables that moderate this relationship, the type of owner and owners' levels of ownership, are considered. The moderating effect of different equity natures on the negative effects of ownership concentration on firm value is analyzed by first classifying these equity natures into SOEs and non-SOEs. The effect of being an SOE or non-SOE on the conflict between majority and minority shareholders is then investigated. The research narrows to a single nation in order to control for heterogeneity in national cultures and political institutions (Hasan et al. 2009). Using a sample of 1265 Chinese listed firms from 2010 to 2016, the hypothesis is tested using panel data techniques-hybrid models augmented with the correlated random effects model.

First, our results confirm a U-shaped non-linear relationship between ownership concentration and Tobin's Q, implying that firm value first decreases and then increases as block holders own more shares. We established that the motivation to expropriate for the entire sample disappears at 58.15% inflexion point. The results mean that at low levels of ownership, the main owner is motivated to extract private benefits because the costs of doing so exceed the benefits. However, as the main owner's stake keeps increasing the costs to extract private benefits overtake the benefits of doing so. The U-shape is consistent with previous research (Wang 2018). Thus we provide support for the research work by Lozano et al. (2016) that generalizations of how ownership affects firm value from previous works based on diffusely held samples cannot be made in an environment with high ownership concentration.

Second, we clarify the important role of ownership as an efficient corporate governance mechanism using the identity of the main shareholder. In particular we examine the effect of different equity natures

i.e., whether equity nature is SOEs or non-SOE. We show how the motivation for extracting private benefits varies between SOEs and non-SOEs and that, SOEs have a higher propensity to expropriate than non-SOEs. Our investigation reveals that the negative effect of ownership concentration is weaker when a firm equity nature is non-state owned enterprises (non-SOEs) compared to state-owned enterprises (SOEs). While ownership concentration appears to be an efficient mechanism for corporate governance its effect is weaker for SOEs compared to non-SOEs. The computed turning points occur at 63.85% and 55.01% for SOEs and non-SOEs respectively. This shows that the negative effect of ownership on firm value is weaker when a company is a non-SOE suggesting that the motivation to expropriate is lower for non-SOEs compared to SOEs.

As proposed by Shleifer and Vishny (1997), our research confirms that concentrated ownership is an essential value-adding element of good corporate governance practices. Contrary to widely dispersed ownership systems where shareholders own an insignificant fraction of outstanding equity, the large equity positions held by block holders effectively give them some control over the firms in which they invest. This study reveals the significant positive role in both SOEs and non-SOEs that ownership concentration plays in enhancing firm value. However the revelation that the propensity to expropriate minority shareholders is less for non-SOEs compared to SOEs suggests that the state's "helping hand" is subordinate to the "grabbing hand." This means that the state being the largest shareholder, its provision for support in terms of financing and resources is outweighed by the conflicts between the two groups. As argued earlier bureaucrats often pursue goals that are completely deviant from SOEs profitability goals, they are contented with achieving their political goals and pursuing any private benefits (Shleifer and Vishny 1997). As noted by Sappington and Stiglitz (1987), the collision of social and political objectives with the firm's profit goals raise difficulties in management monitoring and capital budgeting apart from diluting profit making motives of local governments as corporate controllers.

The research and practice implications of this paper are vast. First, the findings add significant evidence to the existing literature. We expand our knowledge on the implications of ownership concentration in concretizing the value-addition function of good corporate governance practices. Beneficiaries of these results include companies' shareholders, management, foreign investors, regulators, and academics and help in improving the development of the Chinese stock markets. From the results established in this particular research we provide practical guidelines for optimal ownership structure to enhance Chinese SOEs and non-SOEs firm value. With regards to policy we suggest more extensive ownership structure reforms should be undertaken in China to reduce government ownership while promoting private participation.

Second, we provide important implications for practice. We give insight on the behavior of non-SOEs and SOEs according to the level of ownership, which can help minority shareholders to assess appropriate relationships with the various Chinese businesses. To select the most appropriate company to invest, investors should consider distribution of ownership in businesses and the potential for conflicts, which, in turn, potentially increase the chance of expropriation.

While this particular research offers a unique perspective to studying the convenience of ownership as a good mechanism of corporate governance, some limitations should be noted. First, because of the limitations with data issues, we could not delve further to inquire how the relationship of the main shareholder is affected by a second shareholder. In addition the sample is derived from across China, it is therefore expected that provincial differences might influence the firm value ownership relationship. We also recommend that future research should explore alternative mechanisms in particular those external to the firm such as the market for corporate control, the market for managers, and the market for products and services.

**Author Contributions:** Conceptualization, methodology, software, formal analysis, and writing original draft was done by T.F. Supervision, Y.K., writing—review and editing and formal analysis, G.C.-D. and H.S., review and editing, O.K. All authors have read and agreed to the published version of the manuscript.

**Funding:** This research was funded by National Natural Science Foundation of China, grant number 71371081.

**Conflicts of Interest:** The authors declare no conflict of interest.

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
