# Peer review of "Corporate Governance Mechanisms, Ownership and Firm Value: Evidence from Listed Chinese Firms"

_ijfs, doi:10.3390/ijfs8020020_

Round 1
Reviewer 1 Report
The paper has useful and interesting information for the readers.
I have the following remarks for the authors:
- The first sentence of your introduction begins with a minor mistake “The/ In the…” (line 33).
- The paper does not apply the Reference List and Citations Style for MDPI Journals. You need to review/ change your citations according to the recommendations of the Guide.
- The “Introduction” section is far to long. It should briefly place the study in a broad context and highlight controversial and diverging hypotheses when necessary; briefly mention the main aim of the work and highlight the principal conclusions.
- More attention is required between lines 428-435. There are some spelling mistakes.
- The information under each table is no necessary. It can be removed.
- More discussions related to the economic consequences of the research findings are needed in the final section of the paper.
Author Response
Dear Editor/Reviewer
We are grateful for your effort to highlight areas that need improvement in our paper.
Following the first review of our paper titled: Corporate governance mechanisms, ownership and firm value: Evidence from listed Chinese Firms the Reviewer raised the following issues
- The first sentence of your introduction begins with a minor mistake “The/ In the…” (line 33).
- The paper does not apply the Reference List and Citations Style for MDPI Journals. You need to review/ change your citations according to the recommendations of the Guide.
- The “Introduction” section is far too long. It should briefly place the study in a broad context and highlight controversial and diverging hypotheses when necessary; briefly mention the main aim of the work and highlight the principal conclusions.
- More attention is required between lines 428-435. There are some spelling mistakes.
- The information under each table is no necessary. It can be removed.
- More discussions related to the economic consequences of the research findings are needed in the final section of the paper.
We have taken note of these issues and made efforts to address in part or wholly depending on our situation.
- The first sentence was corrected as per recommendation
- We have edited the referencing style in line with the referencing style for MDPI Journals. Citations have been reviewed. In addition we have increased our citations in line with recommendations from the second reviewer to improve on our literature review.
- We have made some changes to our introduction. Though it remains generally long, we have attempted to improve on the presentation of our main argument. We have highlighted our major conclusions from line 124-149.
- Regarding this section we have checked and corrected some words we found to be misspelled.
- We have removed some information underneath the tables that we felt would not make it difficult for readers to interpret the results. Some information was retained though.
- We have also made some changes in the last section of the paper. In line with recommendations from other reviewers, the section was changed to, “Conclusion”.
These changes were specifically addressing reviewer 1 recommendations. However there are some other changes made to the paper that addressed recommendations from the second reviewer.
Reviewer 2 Report
Referee Report
Main Comments and Suggestions
You should clarify the contributions of the paper which are not elaborated well in the current paper. Corporate governance and firm value is a beaten path; in this sense your reference list is too short. You can talk about the following contributions: What insights can you provide based on your finding? Do they push forward our understanding? What should we do with your research? Do you have any suggestions to improve the current regulation or practice? Adding the above discussion and extend your literature review may help you make more contributions and position your contributions better.
The endogeneity problem can be driven by unobservable CEO characteristics you need to discuss. See Coles and Li, 2019. Managerial Attributes, Incentives, and Performance and Coles and Li, 2019.
An Empirical Assessment of Empirical Corporate Finance.
My main suggestion is that you should tell a richer story and link to more literature by discussing more relevant channels. You should consider, for example, market competition as a governance mechanism: Giroud, X., and H., Mueller, 2011, Corporate governance, product market competition, and equity prices. Journal of Finance 66, 563-600. The interactions between the executives, such as mutual monitoring among the executives: Li, Z.F., 2014, Mutual monitoring and corporate governance, Journal of Banking & Finance, 45, 255-269; Li, Z.F., 2018, Mutual monitoring and agency problem. https://www.researchgate.net/publication/272305464_Mutual_Monitoring_and_Agency_Problems; and external interactions between CEOs in the industry tournament: Coles et al. 2018, Industry Tournament Incentives, Review of Financial Studies, 31(4):1418-1459; On inside debt as governance: Li, F., Lin, S., Sun, S., Tucker, A. 2018. Risk-Adjusted Inside Debt. Global Finance Journal 35: 12-42. Or compensation incentives: Core, J. and Guay W., 1999, The use of equity grants to manage optimal equity incentive levels, Journal of Accounting and Economics 28, 151-184.
You need to discuss those aspects of possible channels to give readers a more comprehensive view and a richer story and/or point out future research direction from these perspectives.
Minor Comments and Suggestions
You should study and rationalize the use of firm size measures in the literature since frim size is the key variable in this area and they affect the independent and dependent variables simultaneously. See Dang et al. 2018. Measuring Firm Size in Empirical Corporate Finance. Journal of Banking & Finance, 86:159-176. After all it is the most significant variable in most studies alike. You need to discuss and justify your firm size measure. The results may not be robust to different measures of firm size, which is very common in this area.
There are many typos and grammatical mistakes throughout the paper, making it hard to read and understand. The first sentence of the abstract has mistakes and is not clear. This study analyses the role of ownership as a good corporate governance mechanism. Analyze. Role is a good CG? This study analyzes corporate ownership structure as a corporate governance mechanism and its role in creating firm value.
Try to avoid long sentences and vague words. Use short, precise, and concise sentences and be more straightforward. The last section should be called conclusion where you should summarize all your findings, their implications to researchers and practitioners, future direction for research, limitation of the current study, etc. You need to seriously proofread the paper and extend and update your references.
In conclusion, I would like to thank the authors for a very interesting, unique and potentially important paper. Hope these comments and suggestions can help further their study.
Author Response
Dear Editor/Reviewer
We are grateful for your effort to highlight areas that need improvement in our paper.
Following the first review of our paper titled: Corporate governance mechanisms, ownership and firm value: Evidence from listed Chinese Firms you raised the following issues
- You should clarify the contributions of the paper which are not elaborated well in the current paper. Corporate governance and firm value is a beaten path; in this sense your reference list is too short. You can talk about the following contributions: What insights can you provide based on your finding? Do they push forward our understanding? What should we do with your research? Do you have any suggestions to improve the current regulation or practice? Adding the above discussion and extend your literature review may help you make more contributions and position your contributions better.
- The endogeneity problem can be driven by unobservable CEO characteristics you need to discuss. See Coles and Li, 2019. Managerial Attributes, Incentives, and Performance and Coles and Li, 2019.
- My main suggestion is that you should tell a richer story and link to more literature by discussing more relevant channels. You should consider, for example, market competition as a governance mechanism: Giroud, X., and H., Mueller, 2011, Corporate governance, product market competition, and equity prices. Journal of Finance 66, 563-600. The interactions between the executives, such as mutual monitoring among the executives: Li, Z.F., 2014, Mutual monitoring and corporate governance, Journal of Banking & Finance, 45, 255-269; Li, Z.F., 2018, Mutual monitoring and agency problem. https://www.researchgate.net/publication/272305464_Mutual_Monitoring_and_Agency_Problems; and external interactions between CEOs in the industry tournament: Coles et al. 2018, Industry Tournament Incentives, Review of Financial Studies, 31(4):1418-1459; On inside debt as governance: Li, F., Lin, S., Sun, S., Tucker, A. 2018. Risk-Adjusted Inside Debt. Global Finance Journal 35: 12-42. Or compensation incentives: Core, J. and Guay W., 1999, The use of equity grants to manage optimal equity incentive levels, Journal of Accounting and Economics 28, 151-184.
- You need to discuss those aspects of possible channels to give readers a more comprehensive view and a richer story and/or point out future research direction from these perspectives.
- You should study and rationalize the use of firm size measures in the literature since frim size is the key variable in this area and they affect the independent and dependent variables simultaneously. See Dang et al. 2018. Measuring Firm Size in Empirical Corporate Finance. Journal of Banking & Finance, 86:159-176. After all it is the most significant variable in most studies alike. You need to discuss and justify your firm size measure. The results may not be robust to different measures of firm size, which is very common in this area.
- There are many typos and grammatical mistakes throughout the paper, making it hard to read and understand. The first sentence of the abstract has mistakes and is not clear. This study analyses the role of ownership as a good corporate governance mechanism. Analyze. Role is a good CG? This study analyzes corporate ownership structure as a corporate governance mechanism and its role in creating firm value.
- Try to avoid long sentences and vague words. Use short, precise, and concise sentences and be more straightforward. The last section should be called conclusion where you should summarize all your findings, their implications to researchers and practitioners, future direction for research, limitation of the current study, etc. You need to seriously proofread the paper and extend and update your references.
We have taken note of these issues and made efforts to address in part or wholly depending on our situation.
- We attempted to revisit this aspect in the paper by making our introduction clear, reworking our literature review and highlighting major findings in the conclusion section.
- We took note of this important issue raised. From line 456-481 we attempted to address edogeneity issues by adding more literature to the paper even though we did not specifically use the recommended papers due to time constraints.
- We have made some changes to our introduction. Though it remains generally long, we have attempted to improve on the presentation of our main argument. We have highlighted our major conclusions from line 124-149.
- This was the most interesting and challenging recommendation which called for additional literature review and reworking the paper in line with those recommendations. Most of the papers recommended by the reviewer were found to be very useful. Moreover they also helped unearth more relevant literature to tell an improved story.
- You recommended that we should discuss the Firm Size control variable in more detail. Some papers were recommended which were reviewed and used to add more flesh to the literature. Besides that we also discovered some more papers addressing the issue of firm size that were not among the recommended list.
- We have reviewed the grammar and spellings in this paper to the level we can. We have also corrected the abstract as per recommendations by the reviewer.
- We have followed this recommendation, worked throughout the paper and tried to make our sentences shorter, precise and made our sentences more concise. We have also updated the references and worked on the conclusion section.
These changes were specifically addressing reviewer 2 recommendations. However there are some other changes made to the paper that addressed recommendations from the first reviewer.
Round 2
Reviewer 2 Report
Congratulations on a successful revision